# Emerging Roles of DDB2 in Cancer

**DOI:** 10.3390/ijms20205168

**Published:** 2019-10-18

**Authors:** Pauline Gilson, Guillaume Drouot, Andréa Witz, Jean-Louis Merlin, Philippe Becuwe, Alexandre Harlé

**Affiliations:** 1Institut de Cancérologie de Lorraine, Service de Biopathologie, Université de Lorraine, CNRS UMR 7039 CRAN, 54519 Vandœuvre-lès-Nancy CEDEX, France; p.gilson@nancy.unicancer.fr (P.G.); a.witz@nancy.unicancer.fr (A.W.); jl.merlin@nancy.unicancer.fr (J.-L.M.); 2Faculté des Sciences et Technologies, Université de Lorraine, CNRS UMR 7039 CRAN, 54506 Vandœuvre-lès-Nancy CEDEX, France; guillaume.drouot@orange.fr (G.D.); philippe.becuwe@univ-lorraine.fr (P.B.)

**Keywords:** DDB2, DNA repair, cancers, proliferation, migration, invasion

## Abstract

Damage-specific DNA-binding protein 2 (DDB2) was originally identified as a DNA damage recognition factor that facilitates global genomic nucleotide excision repair (GG-NER) in human cells. DDB2 also contributes to other essential biological processes such as chromatin remodeling, gene transcription, cell cycle regulation, and protein decay. Recently, the potential of DDB2 in the development and progression of various cancers has been described. DDB2 activity occurs at several stages of carcinogenesis including cancer cell proliferation, survival, epithelial to mesenchymal transition, migration and invasion, angiogenesis, and cancer stem cell formation. In this review, we focus on the current state of scientific knowledge regarding DDB2 biological effects in tumor development and the underlying molecular mechanisms. We also provide insights into the clinical consequences of DDB2 activity in cancers.

## 1. Introduction

DDB2 (damage-specific DNA-binding protein 2, also known as p48 subunit) is a 48-kDa protein exclusively localized in the nucleus of mammalian cells. DDB2 is ubiquitously present in human tissues, albeit differentially expressed. High DDB2 expression level has been described in liver, thymus, kidney, and testes, and a low expression level has been described in brain, lung, skin, heart, and muscles [1]. DDB2 is composed of seven WD40 repeat domains and a *N*-terminal helix–loop–helix motif [2]. The WD40 domains play a role as a module for sequence-specific protein–DNA or protein–protein interactions. Both domains appear critical for the biological functions of DDB2.

Numerous overlapping mechanisms have been found to be involved in the regulation of DDB2 expression. DDB2 basal expression is cell cycle-dependent in normal dividing cells with a DDB2 level that gradually increases in the mid-G1 phase and reaches a maximum at the G1/S boundary before dropping in the S-phase [3]. DDB2 transcription was shown to be transiently increased after UV-induced DNA damage in a p53-dependent manner [1]. The p53 protein cooperates with BRCA1 (Breast Cancer Associated protein 1) for its binding to the *ddb2* promoter and inducing *ddb2* transactivation [1] (Figure 1). The TAp63γ (Tumor protein 63 isoform gamma) isoform of the p63 protein that belongs to the p53 family and shares strong structural similarity with p53 is also able to activate DDB2 expression through recognition of the same region upstream of the transcription initiation site [4]. Structural analysis of the *ddb2* promoter shows multiple Sp1 (Transcription factor Sp1) -specific binding sites as usually found in gene promoters with a G–C rich sequence lacking a TATA box and suggesting a critical role for Sp1 for the basal expression of DDB2 [5] (Figure 1). NF1 (Neurofibromin 1) and E2F (Transcription factor E2F) elements are also identified, although probably having a smaller impact on DDB2 regulation [5]. Moreover, DDB2 activity is finely adjusted through post-transcriptional and post-translational mechanisms. An IRES (internal ribosome entry site) element located at the 5′ end of DDB2 mRNA stimulates the translational process of DDB2 in stress conditions such as serum starvation or exposure to doxorubicin [6]. Moreover, the 3′ untranslated region (3′UTR) of DDB2 mRNA harbors an uracil-rich sequence enabling its prompt export to the cytoplasm and thus DDB2 translational upregulation. During the DNA repair process, the Cul4A (Cullin4A) protein regulates DDB2 protein lifespan by means of its ubiquitin ligase functions [3]. The PARylation (Protein poly ADP-ribosylation) of the DDB2 protein in a PARP1-dependent (Poly [ADP-ribose] polymerase 1) manner stabilizes DDB2 and delays its degradation [7]. A PIP (PCNA-interacting protein) box sequence located in the *N*-terminal region of DDB2 enables the interaction between DDB2 and PCNA (proliferating cell nuclear antigen) for DDB2 proteolytic degradation even in the absence of DNA damage [8,9].

DDB2 was originally identified as a component of the UV-DDB human damage-specific DNA-binding heterodimeric complex along with DDB1 (damage-specific DNA-binding protein 1, also named p127 subunit) [2]. The X-ray structure of the UV-DDB complex showed that the interaction between DDB1 and DDB2 is mediated by the contact between the helix–loop–helix domain of DDB2 and two short repetitive β-propeller domains (WD40) of DDB1 [2]. UV-DDB forms a larger complex through the association of the DDB1 adaptor with the CRL ((CUL4A)-Regulator of Cullins-1 (ROC1 or RBX1) E3 ubiquitin ligase) complex [10]. In this structure, DDB1 serves as an adaptor molecule while DDB2 functions as a substrate receptor module that determines the specificity of targeted substrate [10]. DDB2 locally recognizes DNA damage sites and initiates the global genome nucleotide excision repair process (GG-NER) [2]. DDB2 specifically binds to the most frequent UV-damaged DNA lesions including (6-4) pyrimidine–pyrimidone photoproducts (6-4PPs) and cyclobutane pyrimidine dimers (CDPs) and appears essential for the repair of this latter [11]. The WD40 motifs of DDB2 facilitate the access of XPC (Xeroderma Pigmentosum group C) and other repair proteins to DNA lesions through histone modifications and chromatin decondensation that weaken DNA-histones interactions [10,12,13]. DDB2 fulfils this function through the recruitment of poly(ADP-ribose) polymerases (PARP) that append several units of ADP-ribose on histones [7]. As a part of the NER system, DDB2 also contribute to the repair of bulky DNA adducts other than UV-induced ones. For example, DDB2 was shown to recognize many forms of DNA damages, including those induced by platin-derived products, nitrogen mustard, psoralen, abasic sites, as well as single-stranded DNA [14]. However, less is known about the specific role of DDB2 in the repair of these DNA damages.

Beyond its most-studied role in DNA repair pathways, DDB2 appears as a multifunctional protein that participates in other essential biological processes such as gene transcription, cell cycle progression, and protein degradation. Since DDB2 exhibits chromatin-remodeling functions, DDB2 has come to be recognized as a transcription regulator for a wide range of target genes. DDB2 especially participates in the activation of SWI/SNF (SWItch/Sucrose Non Fermenting) remodeling complexes [15] as well as STAGA (SPT3-TAFII31-GCN5L acetylase) and CBP (CREB (cAMP-response-element-binding protein)-binding protein)/p300histone acetyltransferase complexes [16] that contribute to chromatin unfolding and facilitate the recruitment of RNA polymerase to specific regions. DDB2 also stimulates the transcriptional activity of the transcription factor E2F1 and thus the expression of multiple target genes involved in cell cycle progression [3]. As a recognized member of the DDB1 and CUL4-associated factors (DCAF), DDB2 is in a close contact with the E3 ubiquitin ligase of the Cul4A protein and hence participates in the degradation of various proteins such as the cyclin-dependent kinase (CDK) inhibitor p27, a well-known negative regulator of the cell cycle [17]. DDB2 also contributes to the decay of p53 protein to maintain low levels of the CDK inhibitor p21^Waf1/Cip1^ [18].

Several studies reported DDB2 as playing a novel function in the development and progression of various cancers. Herein, we focus on the current state of scientific knowledge regarding the biological effects of DDB2 in tumor development and the underlying molecular mechanisms. We also provide insights into the clinical consequences of DDB2 activity in cancers.

## 2. DDB2: A New Potent Tumor Suppressor?

DDB2 was first considered as a novel tumor suppressor based on the findings that mutations in the ddb2 gene result in an impairment in DDB2–DNA or DDB2–DDB1 interactions and subsequent NER activity defects [2]. Such deficiencies are observed in a subset of Xeroderma pigmentosum hereditary disease (Xeroderoma pigmentosum group E, XP-E) that displays an extreme sensitivity to UV radiation and a high predisposition to skin cancers [7]. Moreover, DDB2-deficient mice are prone to developing a wide panel of tumors, even in the absence of UV exposition [19]. Several studies highlighted an altered DDB2 expression, compared to non-malignant tissues, in many types of cancers [20], including prostate [21], colorectal [22,23], skin [24], head and neck [25], and ovarian [26] cancers. Furthermore, a correlation between low DDB2 expression level and poor outcomes was established among patients with melanoma, colon, ovarian, lung, breast, brain or head, and neck cancers, suggesting a critical role for DDB2 in tumor suppression [22,25,26,27].

## 3. DDB2 Has a Dual Activity on Cancer Cell Proliferation

High levels of DDB2 protein and mRNA are reported in ER (Estrogen receptor)-positive and non-invasive breast cancer models compared to ER-negative aggressive breast cancer cells and mammary non-malignant cells [16]. Enhanced DDB2 expression in DDB2-low level models upregulates in vitro cancer cell growth rate and clonogenicity. Such effects are abrogated by DDB2 knockdown in DDB2-overexpressed breast cancer models, suggesting the oncogenic role of DDB2 in mammary cancer cell growth [16]. DDB2 facilitates cell cycle progression, especially the entry in the S-phase, through the binding of the C-terminal domain and the co-activation of the transcription factor E2F1 [3]. By means of E2F1 transcriptional abilities, DDB2 indirectly regulates the expression of key genes involved in DNA replication and G1/S transition. The stimulating effect of DDB2 on cancer cell growth also involves the downregulation of manganese superoxide dismutase (MnSOD) [16]. MnSOD is a mitochondrial enzyme that detoxifies reactive oxygen species (ROS) to protect cells from oxidative damage. DDB2 interacts with the proximal *SOD2* (Superoxide dismutase 2) promoter, resulting in the loss of H3 histone acetylation, and in the recruitment of the AP-2 transcription factor, which is well known in the repression of the *SOD2* gene and downregulation of the encoded MnSOD [28]. By this mechanism, DDB2 attenuates the elimination of ROS that are known to activate several signaling pathways involved in breast cancer cell growth [28].

In contrast, DDB2 has shown antiproliferative properties in ovarian and prostate cancers in vitro. In an ovarian cancer cell model, DDB2 is reported to negatively regulate NEDD4L (Neural precursor cell expressed developmentally downregulated gene 4-like) cellular levels by inducing histone H3 trimethylation at the *NEDD4L* promoter region [29]. This limits the NEDD4L-dependent proteolytic degradation of the effector proteins SMAD2 (Mothers against decapentaplegic homolog) and SMAD3 and enhances the TGF-β (Transforming growth factor beta) signal transduction downstream, finally contributing to the inhibition of cancer cell proliferation [29].

NRIP/DCAF6 (Nuclear receptor-interacting protein 1/DDB1- and CUL4-associated factor 6) and DDB2 proteins are both members of the DDB1 and CUL4-associated factors DCAF family and androgen receptor (AR)-interacting proteins that physiologically compete to maintain AR expression level. DDB2 mediates the contact between AR and CUL4A–DDB1 E3 ligase complex for AR ubiquitination and proteasomal degradation in a p53-independent manner [30], while NRIP displaces DDB2 from the complex and stabilizes AR [21]. As the DDB2 expression level is found to be lower in prostate cancer tissues compared to non-neoplastic ones, this could interfere with AR homeostasis and induce subsequent AR-dependent prostate cancer growth.

Deregulation in the Wnt signaling pathway is usually co-opted during colon cancer development and is sought to be a driver event in this process. Negative regulatory effects of DDB2 in the Wnt signaling are mediated by the recruitment of β-catenin and the H3K27 methylase EZH2 (Enhancer of zeste homolog 2) to *Rnf43* (Ring Finger protein 43) promoter. This facilitates the activation of the RNF43 enzyme that eliminates Wnt receptors at the cell surface and downregulates Wnt signaling in colorectal cancer cells [31].

## 4. DDB2 Confers Resistance to Radiation and PARP Inhibitors and Sensitizes Cancer Cells to Chemotherapy-Induced Apoptosis

DDB2 participates in cellular responses to radiation-induced DNA damages [32]. Upon ionizing radiations, the DDB2 level transiently increases and facilitates homologous recombination repair of DNA double-strand breaks and prevents apoptosis. This is mediated by the phosphorylation of the G2-arrest-mediating factor Chk1 and confers radioresistance in the NSCLC (Non-small-cell lung carcinoma) cell model. A recent study shows that DDB2 also protects triple negative breast cancer cells from PARP inhibition and apoptosis through the regulation of DNA double-strand break repair by homologous recombination pathway [33]. DDB2 deficiency could thus sensitize TNBC cells to PARP inhibitors.

Given the role of DDB2 in DNA repair pathways, it could be argued that tumors with DDB2 deficiency should be more sensitive to DNA damaging treatments, such as platin products, as they lack some repair. However, DDB2 activity seems not needed to repair cisplatin-induced DNA crosslinks [34]. Besides, DDB2 participates in sensitizing ovarian cancer cells to cisplatin-mediated apoptosis through the downregulation of the *bcl-2* (B-cell lymphoma 2) transcriptional machinery in a HDAC1-dependent (Histone deacetylase 1) manner and the ubiquitylation of the antiapoptotic protein bcl-2 [34].

In the context of NER, DDB2 has a significant role in apoptotic processes through the regulation of p53 activity [35]. Upon exposure to DNA-damaging agents, DDB2-deficient cells fail to undergo apoptosis due to the implication of DDB2 as a member of the UV-DDB complex in the degradation of the CDK (cyclin-dependent kinase) inhibitor p21^Waf1/Cip1^ [20]. In case of DDB2 depletion, p21^Waf1/Cip1^ accumulates in cells associated with a blockade of apoptosis and cell cycle arrest. Cisplatin-resistant Hela cells are found to survive UV-induced apoptosis due to the accumulation of DDB2 associated with the upregulation of the antiapoptotic protein c-Flip (Cellular FLICE-like inhibitory protein) [36].

## 5. DDB2 Influences Epithelial to Mesenchymal Transition (EMT) and Cancer Cell Migration and Invasion

Several studies reported DDB2 as a negative regulatory factor of cancer cell migration and invasion. This notion is supported by the findings that DDB2 associates with the *NFKBIA* (NFKB Inhibitor Alpha) promoter region and enhances the expression of the encoded protein IκBα (nuclear factor of kappa light polypeptide gene enhancer in B-cells inhibitor, alpha), a cytoplasmic inhibitor of NF-κB (nuclear factor κB) [28]. Thus, DDB2 may reduce the NF-κB activity and its associated cancer-invasion effects in breast cancers. DDB2 also modulates nanomechanical properties and stiffness of the mammary cancer cells associated with changes in the cortical actin–cytoskeleton organization and a loss of adhesion capacity [37].

DDB2 also represses head and neck squamous cell carcinoma and colon cancer progression by recruiting the histone H3K9 methylase suv39h (suppressor of variegation 3-9 homolog 1) to limit the transcriptional activity of Snail and ZEB1 (Zinc finger E-box Binding homeobox 1) factors and their epithelial-to-mesenchymal functions [25]. Upregulation of DDB2 levels induces the conversion of colon cancer cells with mesenchymal-type to epithelial-type. Conversely, the depletion of DDB2 in epithelial-type cells induces a morphological change in these cells with the appearance of mesenchymal characteristics. In the same way, DDB2 restricts the transcription of the pro-angiogenic growth factor VEGF (vascular endothelial growth factor) [25] and thus contributes to the inhibition of cancer cell invasion and metastasis formation.

However, another recent study suggests that DDB2 exerts migration and invasion stimulatory functions in gastric cancers. It reveals that DDB2 promotes the ubiquitination and degradation of a newly identified tumor suppressor PAQR3 (progestin and adipoQ receptor family member III) that limits cancer development and progression, notably inhibiting cancer cell migration, invasion, and angiogenesis [38].

## 6. DDB2 Affects Cancer Stem Cell Populations

Some studies reveal the protective role of DDB2 for ovarian cancer progression and recurrence by limiting the cancer stem cell (CSC) population. High levels of DDB2 are shown to halt the formation of ovarian tumor xenografts in vivo [26]. DDB2 exerts this regulatory function through the increase of IκBα protein levels, which results in the repression of the NF-κB signaling pathway and impinges the capacity of CSC to self-renew [26]. DDB2 also represses ovarian cancer cell dedifferentiation by inhibiting the transcription of aldehyde dehydrogenase 1 family member A1 (ALDH1A1) [39]. The impact of DDB2 on cancer stem cell populations is still under investigation in other cancer models and need to be explored.

## 7. Concluding Remarks and Perspectives

DDB2 was originally identified as a sensor of DNA damage that plays a critical role in DNA repair pathways. Sequence-specific interactions of DDB2 with DNAs or proteins make it a multifunctional protein with various functions in chromatin remodeling, gene transcription, cell cycle progression, and protein decay. More recently, the emerging role of DDB2 in the regulation of carcinogenesis has been established. DDB2 was first considered as a new tumor suppressor based on the finding that DDB2 deficiency leads to tumor formation. However, growing evidence about DDB2 activity suggests a regulatory mode more complex than envisioned. Actually, DDB2 seems to have a dual function in cancers, sometimes having tumor-suppressive properties and sometimes functioning as an oncogene. DDB2 exerts multilevel regulation of cancer development and progression, modulating cancer cell growth, migration and invasion, angiogenesis, and CSC formation (Figure 2)

These data suggest that DDB2 may have a potent role as a prognosis and predictive biomarker in cancers. However, they emphasize the need to improve our knowledge on DDB2 functions in the different cancer types, owing to the fact that DDB2 behaves differently according to cancer localization (Figure 3). Considering the importance of DDB2 in cancers, it becomes crucial to better understand its regulation and expression. The potent role of DDB2 as a direct or indirect drug target for the management of patients with cancer needs to be investigated.

## Figures and Tables

**Figure 1 ijms-20-05168-f001:**
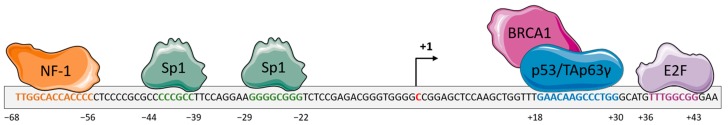
Schematic representation of the regulation of the gene encoding the DDB2 (Damage-specific DNA-binding protein 2) protein. The proximal promoter of *ddb2* gene harbors response elements for the transcription factors NF-1 (Neurofibromin 1) (orange) and Sp1 (Transcription factor Sp1) (green) upstream of the transcription initiation site. The proximal promoter also contains a response element for p53, in association with BRCA1 (Breast cancer type 1 susceptibility protein), or Tap63γ (Tumor protein 63 isoform gamma) (blue) proteins and the E2F (Transcription factor E2F) transcription factor downstream of the transcription initiation site. The binding of these proteins leads to the regulation of *ddb2* gene expression.

**Figure 2 ijms-20-05168-f002:**
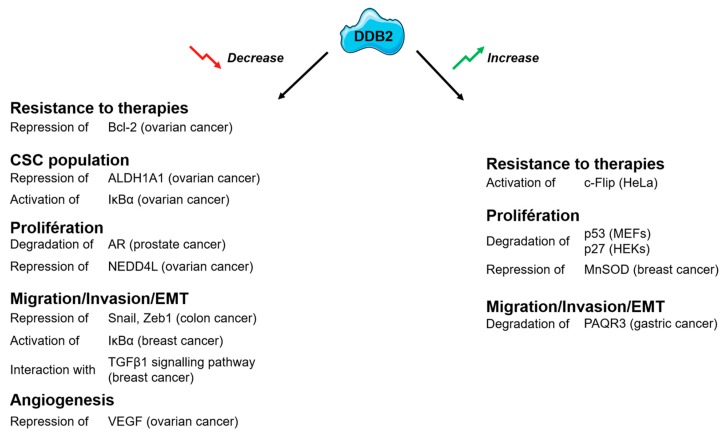
Roles of the DDB2 (Damage-specific DNA-binding protein 2) protein and its identified partners. AR (Androgen Receptor), ALDH1A1 (Aldehyde dehydrogenase 1 family, member A1), Bcl-2 (B-cell lymphoma 2), c-Flip (Cellular FLICE-like inhibitory protein, EMT (Epithelial–mesenchymal transition), HEKs (Human epidermal keratinocytes), IκBα (nuclear factor of kappa light polypeptide gene enhancer in B-cells inhibitor, alpha), MEFs (Mouse embryonic fibroblasts), MnSOD (Manganese superoxide dismutase), NEDD4L (Neural precursor cell expressed developmentally downregulated gene 4-like), PAQR3 (Progestin and adipoQ receptor family member 3), TGF-β1 (Transforming growth factor beta 1), VEGF (vascular endothelial growth factor), Zeb1 (Zinc finger E-box-binding homeobox 1).

**Figure 3 ijms-20-05168-f003:**
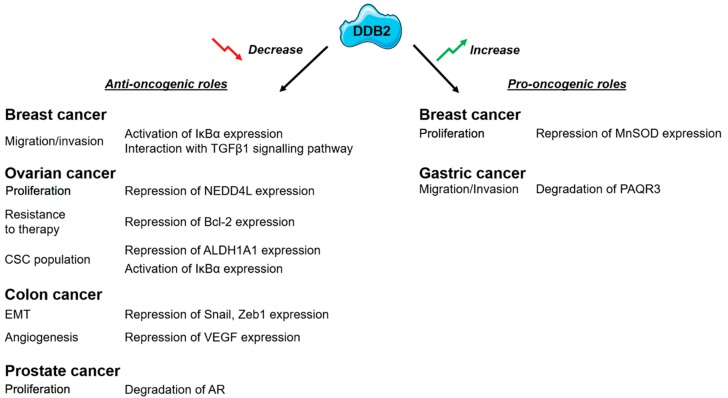
Identified impacts of the DDB2 protein activity in different solid tumors.

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
