# Peer review of "Emerging Roles of DDB2 in Cancer"

_ijms, 2019, doi:10.3390/ijms20205168_

Round 1

Reviewer 1 Report

Nice review, However, I have some questions and remarks.

I have the idea that the DDB2 protein is more involved in the UV damaging pathway than in other DNA damaging repair pathways is this correct or is the UV pathway more studied?

Is the role of DDB2 in cancer stem cells only known for ovarian cancer stem cells?

In line 102 it is written that ‘there is a correlation between low DDB2 expression level and poor outcomes was established among patients suggesting the critical role of DDB2 in tumour suppression’. But are tumors with DDB2 deficiencies not more sensitive to DNA damaging treatments as they lack some repair?

In para 3 an opposite effect of DDB2 is described. Is this only for estrogen receptor positive cells? So I agree with one of the concluding remarks (line 201) that it is crucial to better understand the regulation and expression of DDB2. But can this then be different for all different types of tumors? Or even for similar types of tumors. How do the authors then see DDB2 as a target direct or indirect? Or how could DDB2 be used as a biomarker for treatment (as suggested by Zou et al 2016)?

In lines 142-143 it is stated that ‘Upon ionizing radiations DDB2 facilitates homologous recombination repair of DNA double-strand breaks’. However, in the introduction its stated that ‘DDB2 basal expression is cell cycle-dependent in normal dividing cells with a DDB2 level that gradually increases in mid-G1 phase and drops in S-phase’. How do these two statements fit as HR repair does not (or maybe very, very, rarely) occur in G1 but only in S-phase or G2.

Author Response

Reviewer n°1:

Nice review, However, I have some questions and remarks. I have the idea that the DDB2 protein is more involved in the UV damaging pathway than in other DNA damaging repair pathways is this correct or is the UV pathway more studied?

The DDB2 protein has been widely studied for its role in UV DNA damage repair. However, as a part of the NER system, DDB2 would also contribute to the repair of bulky DNA adducts other than UV-induced ones. For example, DDB2 was shown to recognize many forms of DNA damages including those induced by platin-derived products , nitrogen mustard, N-methyl-N'-nitro-N-nitrosoguanidine, depurination, psoralen, abasic sites as well as single-stranded DNA (Chu and Chang, Science ,1988; Patterson and Chu, Mol Cell Biol, 1989;  Payne and Chu, Mutat Res, 1994; Fujiwara et al., J Biol Chem, 1999; Batty et al., J Mol Biol, 2000; Tang and Chu, DNA repair, 2002). However, less are known about the specific role of DDB2 in the repair of these DNA damages. Zhao et al. notably demonstrated in 2014 that DDB2 was not needed for the repair of cisplatin-induced DNA crosslinks (Zhao et al., Mol. Cancer Res, 2014). The manuscript has been amended accordingly.

Is the role of DDB2 in cancer stem cells only known for ovarian cancer stem cells?

To our knowledge, the role of DDB2 in cancer stem cells has been studied only in ovarian cancer models (Han et al., Mol Cancer Res, 2014; Cui et al., Cell Death Dis., 2018). All the data provided emanate from the same research team. This regulatory function of DDB2 is still under investigation in other cancer models. We have added this notion in the revised manuscript.

In line 102 it is written that ‘there is a correlation between low DDB2 expression level and poor outcomes was established among patients suggesting the critical role of DDB2 in tumour suppression’. But are tumors with DDB2 deficiencies not more sensitive to DNA damaging treatments as they lack some repair?

Thank you for this comment. Although DDB2 is involved in the recognition of DNA damages induced by cisplatin, it seems not needed to repair cisplatin-induced DNA crosslinks. Besides, DDB2 participates in sensitizing ovarian cancer cells to cisplatin-mediated apoptosis through the downregulation of the bcl-2 transcriptional machinery in a HDAC1-dependent manner and the ubiquitylation of the antiapoptotic protein bcl-233.

A recent study shows that DDB2 protects triple negative breast cancer cells from PARP inhibition and interferes with DNA damage accumulation through the regulation of DNA double-strand break repair by homologous recombination pathway (Zhao et al., Cancer Sci, 2019). DDB2 deficiency could thus sensitize TNBC cells to PARP inhibitors. This article has been published at the time of the manuscript submission. We added this reference in the revised manuscript.

In para 3 an opposite effect of DDB2 is described. Is this only for estrogen receptor positive cells? So I agree with one of the concluding remarks (line 201) that it is crucial to better understand the regulation and expression of DDB2. But can this then be different for all different types of tumors? Or even for similar types of tumors. How do the authors then see DDB2 as a target direct or indirect? Or how could DDB2 be used as a biomarker for treatment (as suggested by Zou et al 2016)?

Thank you for this very important question. Kattan et al. show that the introduction of the DDB2 gene into ER-negative cells stimulated growth and colony formation while the knockdown of DDB2 in ER-positive cells inhibited cancer cell growth and colony formation (Kattan et al., PlosOne, 2008).

 It seems that DDB2 functions vary depending on the cancer type as shown in figure 3. In this context, it appears essential to elucidate the role of DDB2 in each specific cancer type before considering it as a potent therapeutic target or biomarker in this pathology.

In lines 142-143 it is stated that ‘Upon ionizing radiations DDB2 facilitates homologous recombination repair of DNA double-strand breaks’. However, in the introduction its stated that ‘DDB2 basal expression is cell cycle-dependent in normal dividing cells with a DDB2 level that gradually increases in mid-G1 phase and drops in S-phase’. How do these two statements fit as HR repair does not (or maybe very, very, rarely) occur in G1 but only in S-phase or G2.

Sorry for the lack of clarity. Upon ionizing radiations, DDB2 transcription is shown transiently increased. DDB2 facilitates homologous recombination repair of DNA double-strand breaks and prevents apoptosis through the phosphorylation of the Chk1 factor and subsequent G2 arrest. The manuscript has been amended accordingly.

Reviewer 2 Report

Comments to the authors

English editing is necessary. Latest references should be checked. In the second paragraph of introduction, logical rearrangement is needed. Besides, it is not appropriated to say the regulation of DDB2 expression are not completed understood. The authors explained a lot about DDB2 regulation in this paragraph. In “2. DDB2: a new potent tumour suppressor?” paragraph, the reference was missing in the first sentence. In the last sentence, the authors should clarify the cancer type. Please divide ascending and descending groups clearly in Figure 2 and 3. The authors should clarify a conclusion section about DDB2.

Author Response

English editing is necessary. Latest references should be checked.

English has been edited accordingly.

A novel article concerning the role of DDB2 in the response to PARP inhibitors has been published at the time of the manuscript submission. This study shows that DDB2 protects triple negative breast cancer cells from PARP inhibition through the regulation of DNA double-strand break repair by homologous recombination pathway (Zhao et al., Cancer Sci, 2019). DDB2 deficiency could thus sensitize TNBC cells to PARP inhibitors. We added this reference in the revised manuscript.

In the second paragraph of introduction, logical rearrangement is needed.

Thank you for this suggestion. Manuscript has been revised accordingly.

Besides, it is not appropriated to say the regulation of DDB2 expression are not completed understood. The authors explained a lot about DDB2 regulation in this paragraph.

We agree with the reviewer’s point of view and modified the sentence in the revised manuscript accordingly.

In “2. DDB2: a new potent tumour suppressor?” paragraph, the reference was missing in the first sentence. In the last sentence, the authors should clarify the cancer type.

Thank you for these suggestions. A reference has been added for the first sentence while the cancer types were detailed in the last sentence.

Please divide ascending and descending groups clearly in Figure 2 and 3.

Thank you for pointing this out. We modified the figures accordingly.

The authors should clarify a conclusion section about DDB2.

Conclusion has been clarified accordingly.

Reviewer 3 Report

The manuscript "Emerging roles of DDB2 in cancer" is a timely review of the recent discoveries involving DDB2 protein roles in cancer. Overall the paper is very well written, with clear and concise figures summarizing sometimes contradictory findings on the DDB2 participation in various cancers.
Some noticed typos and minor issues:
lines 34 and 42 - "figure 1" should start with capital
line 132-134 - the sentence is unclear, should be rephrased.

Author Response

The manuscript "Emerging roles of DDB2 in cancer" is a timely review of the recent discoveries involving DDB2 protein roles in cancer. Overall the paper is very well written, with clear and concise figures summarizing sometimes contradictory findings on the DDB2 participation in various cancers.
Some noticed typos and minor issues:

lines 34 and 42 - "figure 1" should start with capital
line 132-134 - the sentence is unclear, should be rephrased.

Thank you for your positive feedback and suggestions. the manuscript has been revised accordingly.

Reviewer 4 Report

This review entitled “Emerging roles of DDB2 in cancer” describes the recent progress towards understanding the current state of scientific knowledge regarding DDB2 biological effects in tumour development and the underlying molecular mechanisms. It provides insights into the clinical consequences of DDB2 activity in cancers. Authors detail an overview about the functions, mechanism of DDB2 in the regulation of potent tumour suppressor; cancer cell proliferation; radioresistance and sensitizes cancer cells to chemotherapy-induced apoptosis; epithelial to mesenchymal transition (EMT) and the cancer cells migration and invasion; cancer stem cells population. DDB2 exerts a multilevel regulation of cancer development and progression, modulating cancer cell growth, migration and invasion, angiogenesis and CSC formation and behaves differently according to cancer localization. It suggests that DDB2 may have a potent role as prognosis and predictive biomarker in cancers. Considering the importance of DDB2 in cancers, it becomes crucial to better understand its regulation and expression. The potent role of DDB2 as a direct or indirect drug target for the management of patients with cancer needs to be investigated.

Author Response

This review entitled “Emerging roles of DDB2 in cancer” describes the recent progress towards understanding the current state of scientific knowledge regarding DDB2 biological effects in tumour development and the underlying molecular mechanisms. It provides insights into the clinical consequences of DDB2 activity in cancers. Authors detail an overview about the functions, mechanism of DDB2 in the regulation of potent tumour suppressor; cancer cell proliferation; radioresistance and sensitizes cancer cells to chemotherapy-induced apoptosis; epithelial to mesenchymal transition (EMT) and the cancer cells migration and invasion; cancer stem cells population. DDB2 exerts a multilevel regulation of cancer development and progression, modulating cancer cell growth, migration and invasion, angiogenesis and CSC formation and behaves differently according to cancer localization. It suggests that DDB2 may have a potent role as prognosis and predictive biomarker in cancers. Considering the importance of DDB2 in cancers, it becomes crucial to better understand its regulation and expression. The potent role of DDB2 as a direct or indirect drug target for the management of patients with cancer needs to be investigated.

Thank you for taking the time to review our manuscript and for your positive feedback.

Round 2

Reviewer 2 Report

No